A novel control of human keratin expression: cannabinoid receptor 1-mediated signaling down-regulates the expression of keratins K6 and K16 in human keratinocytes in vitro and in situ

Ramot Yuval 1 2
Sugawara Koji 1 3
Zákány Nóra 1 4
Tóth Balázs I. 4 5
Bíró Tamás 4
Paus Ralf ralf.paus@uksh.de 1 6
1 Department of Dermatology, University of Luebeck , Luebeck , Germany
2 Department of Dermatology, Hadassah-Hebrew University Medical Center , Jerusalem , Israel
3 Department of Dermatology, Osaka City University Graduate School of Medicine , Osaka , Japan
4 DE-MTA “Lendület” Cellular Physiology Research Group, Department of Physiology, MHSC, RCMM, University of Debrecen , Debrecen , Hungary
5 Laboratory of Ion Channel Research and TRP Research Platform Leuven (TRPLe), Department of Cellular and Molecular Medicine, KU Leuven , Leuven , Belgium
6 Institute of Inflammation and Repair, and Dermatology Centre, University of Manchester , Manchester , UK
Dollé Pascal
Electronic publication date: 2013 Feb 19
Publication date: 2013
Volume: 1
Electronic Location ID: e40
Received 2012 Nov 30; Accepted 2013 Jan 28
Copyright: © 2013 Ramot et al.
Copyright year: 2013
Copyright holder: Ramot et al.
License: This is an open access article distributed under the terms of the Creative Commons Attribution License, which permits unrestricted use, distribution, and reproduction in any medium, provided the original author and source are credited.
License URL: https://creativecommons.org/licenses/by/3.0/

Keywords: Cannabinoid, Keratin, Psoriasis, Wound healing

Funding: Daniel Turnberg United Kingdom/Middle East Travel Fellowship Scheme Osaka City University “Lendület” grant of the Hungarian Academy of Sciences University of Luebeck This study was supported in part by the Daniel Turnberg United Kingdom/Middle East Travel Fellowship Scheme, administered by the Academy of Medical Sciences to YR; a faculty grant from Osaka City University to KS; a “Lendület” grant of the Hungarian Academy of Sciences to TB; and a faculty grant from the University of Luebeck to RP. The funders had no role in study design, data collection and analysis, decision to publish, or preparation of the manuscript.

==============================
Cannabinoid receptors (CB) are expressed throughout human skin epithelium. CB1 activation inhibits human hair growth and decreases proliferation of epidermal keratinocytes. Since psoriasis is a chronic hyperproliferative, inflammatory skin disease, it is conceivable that the therapeutic modulation of CB signaling, which can inhibit both proliferation and inflammation, could win a place in future psoriasis management. Given that psoriasis is characterized by up-regulation of keratins K6 and K16, we have investigated whether CB1 stimulation modulates their expression in human epidermis. Treatment of organ-cultured human skin with the CB1-specific agonist, arachidonoyl-chloro-ethanolamide (ACEA), decreased K6 and K16 staining intensity in situ. At the gene and protein levels, ACEA also decreased K6 expression of cultured HaCaT keratinocytes, which show some similarities to psoriatic keratinocytes. These effects were partly antagonized by the CB1-specific antagonist, AM251. While CB1-mediated signaling also significantly inhibited human epidermal keratinocyte proliferation in situ, as shown by K6/Ki-67-double immunofluorescence, the inhibitory effect of ACEA on K6 expression in situ was independent of its anti-proliferative effect. Given recent appreciation of the role of K6 as a functionally important protein that regulates epithelial wound healing in mice, it is conceivable that the novel CB1-mediated regulation of keratin 6/16 revealed here also is relevant to wound healing. Taken together, our results suggest that cannabinoids and their receptors constitute a novel, clinically relevant control element of human K6 and K16 expression.

Introduction

Endocannabinoids as well as exocannabinoids (such as the active components of cannabis) control the function of various types of cells via cannabinoid receptor (CB)-dependent or independent manner (Kupczyk, Reich & Szepietowski, 2009). The endocannabinoid system (ECS) consists of these CBs, their endogenous ligands (i.e. endocannabinoids, such as anandamide [AEA] and 2-arachidonoylglycerol), and enzymes responsible for endocannabinoid synthesis and degradation (Biro et al., 2009). In human skin, many different types of cells are now known to express functional CBs (Biro et al., 2009; Czifra et al., 2012; Pucci et al., 2012; Roelandt et al., 2012; Stander et al., 2005; Sugawara et al., 2012; Telek et al., 2007; Toth et al., 2011). The ECS is increasingly appreciated as an important regulator of skin function in health and disease. For example, the ECS has become implicated in pain (Khasabova et al., 2012; Walker & Hohmann, 2005) and itch control (Stander, Reinhardt & Luger, 2006), and the modulation of inflammation (Klein, 2005) and allergy (Karsak et al., 2007). In addition, CB1 signaling is important in mast cell activation and intracutaneous mast cell maturation from resident progenitors (Sugawara et al., 2012). Furthermore, it regulates fibrosis (Akhmetshina et al., 2009), sebocyte differentiation (Dobrosi et al., 2008) and eccrine epithelial biology (Czifra et al., 2012). Nevertheless, the functions of CB-mediated signaling in human keratinocytes (KCs) in situ are as yet poorly understood.

We have previously shown that outer root sheath KCs of human hair follicles (HFs) express CB1. CB1 stimulation by the endocannabinoid, AEA, markedly inhibited human HF growth by inhibiting hair matrix KC proliferation and inducing apoptosis, thus leading to premature HF involution (catagen development). This was reversed by the CB1-specific antagonist, AM251 (Telek et al., 2007). Similarly, human epidermal KC express CBs, and their differentiation is regulated via CB1 (Maccarrone et al., 2003; Paradisi et al., 2008). AEA also markedly suppresses human epidermal KC proliferation and induces apoptosis via CB1 in vitro and in situ (Toth et al., 2011). This suggests that the ECS could become a useful therapeutic target in the management of chronic hyperproliferative human skin diseases, such as psoriasis (Toth et al., 2011).

However, it remains unclear whether and how CB1-mediated signaling impacts on human KC differentiation, namely on the expression of hyperproliferation-associated keratins. Psoriasis is characterized by the upregulation of keratins K6 and K16 expression within lesional epidermis (Korver et al., 2006; Mommers et al., 2000). This pair of keratins is also prominently up-regulated in the epidermis under wound healing conditions in men and mice (Moll, Divo & Langbein, 2008; Paladini et al., 1996; Rotty & Coulombe, 2012) and is constitutively expressed in the outer root sheath KCs of human HFs (Langbein & Schweizer, 2005; Moll, Divo & Langbein, 2008; Ramot et al., 2009). Psoriasis is a chronic inflammatory, hyperproliferative dermatosis that, in addition to its anti-proliferative properties (Telek et al., 2007; Van Dross et al., 2012), might also profit from the well-recognized anti-inflammatory properties of CB1-mediated signaling (Sugawara et al., 2012; Wilkinson & Williamson, 2007). Therefore, we have investigated whether CB1 stimulation modulates K6 and K16 expression in human skin. This question was made particularly interesting in view of the most recent discovery that, in murine skin, K6 is not only a wound healing-associated keratin, but actively down-regulates KC migration during wound repair (Rotty & Coulombe, 2012).

In order to answer this question, we used the CB1-specific agonist, arachidonoyl-chloro-ethanolamide (ACEA) (Harvey et al., 2012), and checked its effect on K6 expression in situ. This was done by utilizing full thickness human skin organ culture (Lu et al., 2007) as a physiologically and clinically relevant model to study multiple aspects of human skin biology under clinically relevant conditions in vitro (Bodo et al., 2010; Knuever et al., 2012; Langan et al., 2010; Lu et al., 2007; Sugawara et al., 2012). In order to confirm the CB1-specificity of the observed effects of ACEA, we also used the CB1-specific antagonist, AM251 (Chanda et al., 2011).

K16 serves as the type I keratin partner of K6 in the formation of intermediate filament heterodimers (Moll, Divo & Langbein, 2008). It is also involved in epidermal barrier function (Grzanka et al., 2012; Thakoersing et al., 2012), and is up-regulated in hyperprolifeative conditions of the skin such as psoriasis (Iizuka et al., 2004) and atopic dermatitis (Grzanka et al., 2012). Therefore, we also examined the effects of ACEA on K16 expression.

HaCaT cells are a highly proliferating human KC line known to overexpress K6 (Ryle et al., 1989). Since HaCaT KCs share some other characteristics with psoriatic KCs and are often employed as surrogate “psoriatic” KCs (Balato et al., 2012; Farkas et al., 2001; George et al., 2010; Kim et al., 2011; Ronpirin & Tencomnao, 2012; Saelee, Thongrakard & Tencomnao, 2011), we also tested whether and how ACEA modulated K6 expression in these cells in vitro. In order to delineate whether any such effects on keratin expression resulted only indirectly from a possible down-regulation of KC proliferation by CB1 stimulation (Toth et al., 2011), double-labeling and quantitative immunohistomorphometry for both K6 and Ki-67 was performed. Finally, to investigate whether K6-expressing human KCs co-express CB1, double-immunolabeling for both antigens was employed.

Materials and methods

Human skin organ culture

Isolated human skin samples obtained from elective plastic surgery procedures (32 pieces of skin fragments obtained by 4 mm punch biopsies from 4 individuals; 3 females and a male aged 26–74, average: 56.5; 3 skin samples were taken from the scalp and one was taken from the hip) were maintained in supplemented serum-free William’s E medium as previously reported (Bodo et al., 2010; Knuever et al., 2012; Lu et al., 2007; Poeggeler et al., 2010). Human tissue collection and handling was performed according to Helsinki guidelines, after institutional review board ethics approval (University of Luebeck) and informed patient consent.

Skin samples were first incubated overnight to adapt to culture conditions after which the medium was replaced and vehicle or test substances were added. For human skin organ culture, skin samples were treated with ACEA (Sigma-Aldrich, Taufkirchen, Germany, 30 µM) or AM251 (Sigma-Aldrich, 1 µM), or the combination of them for 1-day after the overnight incubation (Sugawara et al., 2012). Following culturing for the time indicated, skin samples were cryoembedded and prepared for histology, immunohistochemistry/immunofluorescence and quantitative immunohistomorphometry (Ramot et al., 2010; Ramot et al., 2011; Sugawara et al., 2012). Each evaluation was performed on 2–4 sections of 2 skin fragments per each treatment group from 2–4 individuals.

Cell culture

Human immortalized HaCaT KCs (Boukamp et al., 1988) were cultured in DMEM (Sigma-Aldrich) supplemented with 10% fetal bovine serum (Invitrogen, Paisley, UK) and antibiotics (PAA Laboratories, Pasching, Austria). For qRT-PCR, the cells were cultured with ACEA (1 µM) for 8 h.

qRT-PCR

qRT-PCR was performed on an ABI Prism 7000 sequence detection system (Applied Biosystems/Life Technologies, Foster City, CA, USA) using the 5’ nuclease assay as detailed in our previous reports (Toth et al., 2011; Toth et al., 2009). Total RNA was isolated from HaCaT keratinocytes using TRIreagent (Applied Biosystems/Life Technologies, Foster City, CA, USA) and digested with recombinant RNase-free DNase-1 (Applied Biosystems) according to the manufacturer’s protocol. After isolation, one µg of total RNA was reverse-transcribed into cDNA by using High Capacity cDNA kit (Applied Biosystems) following the manufacturer’s protocol. PCR amplification was performed by using specific TaqMan primer and probes (Applied Biosystems, assay ID: Hs01699178_g1 for human K6A). As internal housekeeping gene control, transcripts of cyclophillin A (PPIA) were determined (Assay ID: Hs99999904 for human PPIA). The amount of the K6A transcripts was normalized to the control gene using the ΔCT method.

Immunohistochemistry

For the detection of K6 in organ cultured human skin as well as cultured HaCaT KCs, indirect immunofluorescence staining was performed using mouse anti-human K6 antibody (Progen, Ks6.KA12) at 1:10 dilution as a primary antibody and rhodamine conjugated goat anti-mouse IgG (Jackson Immunoresearch Laboratories, West Grove, PA) at 1:200 dilution in phosphate-buffered saline (PBS) as a secondary antibody.

To study the proliferation of epidermal KCs, double-immunostaining for K6 and Ki-67 was performed. Briefly, after the staining for K6 with FITC conjugated goat anti-mouse IgG (Jackson Immunoresearch Laboratories) as a secondary antibody, sections were incubated overnight at 4 °C with a mouse anti-human Ki-67 antibody (DAKO, Hamburg, Germany) at 1:20 in PBS. Sections were then washed with PBS, followed by incubation with rhodamine conjugated goat anti-mouse IgG (Jackson Immunoresearch Laboratories) (1:200 in PBS, 45 min) at room temperature.

To investigate the localization of CB1 and K6, double immunostaining was performed. For CB1 immunostaining, the highly sensitive tyramide signal amplification (TSA) technique (Perkin Elmer, Boston, MA) was applied. Cryosections were incubated overnight at 4 °C with rabbit anti-CB1 (Santa Cruz, CA, USA) at 1:400 diluted in TNB (Tris, NaOH, Blocking reagent, TSA kit; Perkin-Elmer). Thereafter, the samples were labeled with goat biotinylated antibody against rabbit IgG (Jackson Immunoresearch Laboratories) at 1:200 in TNB for 45 min at room temperature. Sections were then stained with streptavidin-conjugated horseradish peroxidase (1:100, 30 min, TSA kit) and were finally incubated with rhodamine conjugated tyramide (1:50, TSA kit). The TSA method was applied according to the manufacturer’s protocol. For the second primary labeling, mouse anti-human K6 antibody (Progen) was applied at 1:20 in PBS, overnight at 4 °C. After the wash with PBS, the sections were incubated with FITC conjugated goat anti-mouse IgG (Jackson Immunoresearch Laboratories) (1:200 in PBS, 45 min) at room temperature.

For K16 antigen detection, we used the LSAB (DCS, Germany) detection method (Ramot et al., 2009) using the K16-gp as primary antibody (guinea-pig, PROGEN, Heidelberg, Germany, dilution 1:1000, GP-CK16), and biotinylated goat anti-guinea pig as secondary antibody (Vector Laboratories, Burlingame, CA, USA, dilution 1:200).  HistoGreen (Linaris, Wertheim-Bettingen, Germany) was used as peroxidase substrate.

For all immunostainings, the respective primary antibodies were omitted as negative controls, and morphological criteria and reproduction of the previously published intracutaneous expression patterns of the examined antigens were used as internal positive and negative controls (Moll, Divo & Langbein, 2008). For all experiments, control and treated sections were stained (and later evaluated) on the same day by the same investigator. To avoid staining biases, we calculated the relative staining intensity (arbitrary intensity; 1 as control group) among treatment groups per each individual and then pooled data from all of the experiments.

High magnification images of K6/Ki67 double immunofluorescence and K6 immunofluorescence on HaCaT cells were taken by laser scanning confocal microscopy (Fluoview 300, Olympus Tokyo, Japan) running Fluoview 2.1 software (Olympus).

The staining intensity of K6 and K16 in defined reference areas was assessed by quantitative immunohistomorphometry using the ImageJ software (NIH: National Institutes of Health, Bethesda, MD) (Bodo et al., 2010; Ramot et al., 2010; Ramot et al., 2011). For epidermal evaluation, staining intensity was evaluated in the suprabasal cells, and for HaCaT cells, staining intensity was measured in the colonies formed. For K6 immunofluorescence intensity, skin sections from 4 different individuals were used, while K16 immunofluorescence intensity and %Ki67 were evaluated in skin sections from 2 patients. 4–6 sections per one individual (2 sections per investigated skin fragment) were used for each evaluation. Each section was evaluated in two/three different non-adjacent microscopic fields (×200), and the mean intensity was measured, and considered as a value. Each treatment group was compared to the control group (average value), and relative change in expression was calculated. Highly comparable results were obtained from different sections from different individuals.

Statistical analysis

Significance of difference between two groups was evaluated using Student’s t-test for unpaired samples. For multiple comparisons, one-way analysis of variance (ANOVA) was used, followed by Bonferroni’s multiple comparison test, using Prism 5.0 software (GraphPad Prism Program, GraphPad, San Diego, CA). p values <0.05 were regarded as significant. All data in the figures are expressed as mean + SEM. *p < 0.05, **p < 0.01, ***p < 0.001 for the indicated comparisons.

Results and discussion

The CB1-selective agonist, ACEA, down-regulates K6 protein expression in situ

First, we asked whether the CB1-specific synthetic agonist ACEA (Pertwee et al., 2010; Sugawara et al., 2012) can modulate the expression of keratin K6 in human skin. K6 staining intensity within the epidermis of full-thickness human skin that had been organ-cultured for 24 h under serum-free conditions in the presence of ACEA (30 µM) or vehicle alone was assessed by quantitative immunohistomorphometry. This showed that K6 immunoreactivity (IR) was significantly reduced after ACEA treatment, compared to the vehicle control group (Figs. 1A and 1B).

Figure 1 The CB1 specific agonist, ACEA significantly inhibits K6 and K16 expression in situ.

(A) Representative images of K6 immunofluorescence with organ cultured human skin treated with ACEA/AM251 (1-day). (B) Statistical analysis of K6 immunofluorescence intensity in organ cultured human skin (quantitative immunohistomorphemtry, ImageJ); stimulation with ACEA (30 µM), AM251 (1 µM) or both for 1-day. n = 9–22 skin sections/group. (C) Representative images of K16 immunohistochemistry with organ cultured human skin samples with ACEA (1-day). (D) Quantitative K16 immunohistomorphometry within the epidermis of organ-cultured human skin samples after 1-day of stimulation with ACEA (30 µM). n = 4 skin sections/group. Data are expressed as mean + SEM. *p < 0.05, **p < 0.01.

This down-regulation was abrogated in part by the co-administration with the CB1-specific antagonist, AM251 (Pertwee et al., 2010; Sugawara et al., 2012) (Figs. 1A and 1B). Therefore, intraepidermal K6 protein expression in normal human skin in situ is down-regulated in a CB1-specific manner.

ACEA also down-regulates K16 protein expression in situ

Since K16 is the type I keratin partner of K6 in KCs and is thought to stabilize this keratin protein as a cytoskeletal heteropolymeric intermediate filament (Ramot et al., 2009), we next analyzed K16 IR in the epidermis of organ cultured human skin samples treated with ACEA. In line with the K6 protein expression, K16 IR was also significantly down-regulated by ACEA in situ (Figs. 1C and 1D).

CB1-mediated signaling also regulates K6 expression in cultured, hyperproliferative human keratinocytes

In order to check whether the observed CB1-mediated effects on K6 regulation within intact human skin epithelium depend on intact epithelial-mesenchymal interactions between epidermis and dermis, or are likely to reflect a direct impact of CB1 ligands on epidermal keratinocytes, we next investigated K6 expression in cultured human HaCaT KCs. This transformed human KC line is well-appreciated to constitutively express K6 and to be hyperproliferative (just like human wounded and psoriatic KC) (Balato et al., 2012; Farkas et al., 2001; George et al., 2010; Kim et al., 2011; Ronpirin & Tencomnao, 2012; Ryle et al., 1989; Saelee, Thongrakard & Tencomnao, 2011). K6 is expressed in hyper-proliferative cells (Weiss, Eichner & Sun, 1984) and both K6 expression and basal layer epidermal KC proliferation are increased in psoriasis lesions (Donetti et al., 2012; Griffiths & Barker, 2007; Korver et al., 2006; Litvinov et al., 2011; Mommers et al., 2000). HaCaT cells are known to express functional CB1 and CB2 (Leonti et al., 2010; Maccarrone et al., 2003; Paradisi et al., 2008), and this had been confirmed previously by our group, both on the gene (RT-PCR analysis) and protein levels (immunocytochemistry and western blotting techniques) (Toth et al., 2011). Thus, being a direct target of CB1-mediated signaling, this makes these KCs not only an instructive cell culture tool for evaluating the direct, dermis-independent role of CB1-mediated signaling in the regulation of keratin expression in human KCs, but may also provide first indications as to how the observed K6 expression could relate to wound healing and/or psoriasis.

In accordance with our human skin organ culture results, ACEA (1 µM) significantly down-regulated K6 staining intensity of HaCaT cells in vitro (Figs. 2A and 2B). This was abrogated by the co-administration of the selective CB1 antagonist, AM251 (100 nM) (Figs. 2A and 2B). Unexpectedly, though, AM251 alone had a partial inhibitory effect on K6 expression, although not significant. This may be related to the fact that AM251 is an inverse agonist (Dono & Currie, 2012; Fiori et al., 2011), and invites further study. The inhibitory effect of ACEA on K6 protein expression was further confirmed by quantitative RT-PCR (Fig. 2C). Therefore, while HaCaT cells may exhibit relatively low CB expression levels, at least under our assay conditions, they showed a vigorous response to CB ligands.

Figure 2 The CB1 specific agonist, ACEA significantly inhibits K6 expression in cultured HaCaT cells.

(A) Representative images of K6 immunofluorescence of cultured HaCaT KCs with ACEA (1 µM), AM251 (100 nM) or both for 1-day. (B) Statistical analysis of K6 immunofluorescence intensity of cultured HaCaT cells. n = 6 colonies/group (C) Statistical analysis of K6 gene expression in HaCaT cells treated with vehicle control or ACEA (1 µM) for 8 h. Data are expressed as mean + SEM. *p < 0.05; **p < 0.01; ***p < 0.001.

The CB1 specific agonist, ACEA significantly decreases human epidermal keratinocyte proliferation in situ

However, the observed effects of CB1-mediated signaling on epidermal K6 expression could simply reflect the appreciated anti-proliferative effects of CB1 agonists (Casanova et al., 2003; Hermanson & Marnett, 2011; Toth et al., 2011). Moreover, K6 is overexpressed in hyper-proliferative and wounded KCs (Weiss, Eichner & Sun, 1984), and both K6 expression and basal KC proliferation are increased in psoriatic epidermal lesions (Donetti et al., 2012; Griffiths & Barker, 2007; Korver et al., 2006; Litvinov et al., 2011; Mommers et al., 2000; Navarro, Casatorres & Jorcano, 1995). Therefore, we next assessed whether CB1 stimulation by CB1 specific agonist, ACEA, could affect human KC proliferation in situ.

Just as we had seen before with the non-selective endocannabinoid, AEA (Toth et al., 2011), the CB1-specific synthetic agonist, ACEA indeed significantly decreased human epidermal KC proliferation in situ. This effect was abrogated by the CB1-specific antagonist, AM251 (assessed by quantitative Ki-67 immunomorphometry, Figs. 3A and 3B).

Figure 3 The CB1 specific agonist, ACEA significantly decreases human epidermal keratinocyte proliferation in situ.

(A) Representative images of K6 (green) and Ki67 (red) double-immunofluorescence with organ cultured human skin treated with ACEA/AM251 (1-day). (B) Quantitative analysis of the percentage of Ki67 + KCs within organ cultured human epidermis. *p < 0.05; ***p < 0.001. n = 5–12 skin sections/group.

The CB1 specific agonist, ACEA, significantly decreases K6 expression in suprabasal cells in a proliferation-independent manner

Therefore, it needed to be dissected whether or not CB1 also reduces K6 expression in a proliferation-independent manner. This was done by selectively assessing K6 expression in non-proliferative (i.e. Ki-67-negative) epidermal KCs in situ. We found that K6 IR within non-proliferative epidermis was also reduced by ACEA (Figs. 4A and 4B). Furthermore, K6-expressing cells in the epidermis co-expressed CB1 in situ (Fig. 4C), suggesting a direct effect of CB1-agnosits on K6-expressing human epidermal KCs in situ. Thus, CB1 stimulation may affect K6 expression both, by reducing KC proliferation and by down-regulating K6 expression directly via CB1 in a proliferation-independent manner.

Figure 4 The CB1 specific agonist, ACEA, significantly decreases K6 expression in suprabasal cells in a proliferation-independent manner.

(A) Representative images of K6 (green) and Ki-67 (red) double immunofluorescence. Dotted rectangles indicate the reference area for quantitative immunohistomorphometry of K6 fluorescence intensity. (B) Quantitative analysis of K6 fluorescence intensity in non-proliferating (i.e. Ki67-negative) cells within human epidermis in situ. Data are expressed as mean + SEM. *p < 0.05. n = 5–7 skin sections/group. (C) K6 (green) and CB1 (red) double-immunofluorescence study. Yellow arrows denote double-positive KCs.

Here we provide the first evidence that CB1-mediated signaling directly regulates K6/16 expression within normal human skin. Specifically, we show that CB1 stimulation down-regulates expression of the hyper-proliferation-associated human keratins K6 in vitro and in situ, and inhibits human epidermal KC proliferation in situ.

The effect of CB-mediated signaling in human KC biology remains to be fully explored. As we have also observed in isolated human skin (Fig. 4C), CB1 protein expression is detected mainly above the basal layer of the epidermis (Stander et al., 2005), i.e. above the compartment where KC proliferation normally occurs most prominently. Wilkinson and Williamson reported that the non-selective CB agonist HU210 inhibited KC proliferation. However, this could not be blocked by either CB1 or CB2 antagonists, suggesting that cannabinoids may also inhibit human KC proliferation through a non-CB1/CB2 mechanism (Wilkinson & Williamson, 2007). Nevertheless, it has been previously shown that AEA, which can interact with CB1 on human KC (Biro et al., 2009), inhibited human KC proliferation in situ and in vitro (Toth et al., 2011).

Therefore, it was important to clarify whether specific CB1 stimulation inhibits human epidermal KC proliferation in situ. By using CB1-specific agonists and antagonists we clearly demonstrate that exclusive CB1 stimulation inhibited KC proliferation. Thus, CB1 is an important key regulator of human KC proliferation. Given the role of epidermal hyperproliferation in the pathobiology of psoriasis (Donetti et al., 2012; Griffiths & Barker, 2007; Korver et al., 2006; Litvinov et al., 2011; Mommers et al., 2000; Navarro, Casatorres & Jorcano, 1995), cannabimimetic agents that activate CB1, therefore, deserve consideration as a novel pharmacological strategy for treating psoriasis.

Furthermore, increased numbers of activated mast cells are often observed in and around lesional psoriatic skin, and increasing evidence suggests that mast cells are functionally important key immunocytes in the pathogenesis of psoriasis (Carvalho, Nilsson & Harvima, 2010; Namazi, 2005; Radosa et al., 2011; Suttle et al., 2012; Toruniowa & Jablonska, 1988). Recently, we have shown that CB1 activation limits excessive mast cell activity and even inhibits mast cell maturation of resident, intracutaneous progenitors (Sugawara et al., 2012). Therefore, besides their anti-proliferative effects on human epidermal KCs, the anti-inflammatory (Richardson, Kilo & Hargreaves, 1998) and mast cell-inhibitory properties of CB1 agonists in human skin (Sugawara et al., 2012) make them a particularly attractive class of agents in future psoriasis management.

It should be noted, that the constitutive level of K6 expression in organ-cultured human skin fragments is considerably higher than normal scalp skin in vivo. Presumably, this occurs as a response to tissue dissection and organ culture, which is well-known to elicit an immediate wound healing response in the epithelium. The latter rapidly starts to migrate over the wound edge in an attempt to enwrap the exposed skin mesenchyme (epiboly phenomenon Stenn, 1981; Brown et al., 1991). This constitutive up-regulation of K6 in organ-cultured normal human skin may greatly heighten the sensitivity to K6 expression-modulatory compounds, such as CB ligands, thus making human skin organ culture a particularly sensitive and instructive tool for clinically relevant keratin research. At the same time, however, caution is advised in extrapolating from these observation made in what essentially reflects a wound healing milieu to healthy, unmanipulated human skin in vivo.

The current findings invite the speculation that the therapeutic down-modulation of K6 and/or K16 expression by CB1 agonists and other cannabimimetics might become exploitable for the management of other dermatoses besides psoriasis, for example pachyonychia congenita (Hickerson et al., 2011; Zhao et al., 2011) and acne (Biro et al., 2009), and could be used to modulate KC migration-dependent reepithelialization in wound healing, similar to related findings in periodontal and intestinal wound repair (Kozono et al., 2010; Wright et al., 2005).

Conclusion

Our results suggest that cannabinoids and their receptors constitute a novel, clinically relevant control element of human K6 and K16 expression. Therefore, cannabimimetic agents might be relevant for the treatment of several skin conditions related to aberrant K6/K16 expression, such as psoriasis and wound healing. In addition, skin organ culture is shown to be a clinically and physiologically relevant model system for investigating the effect of CB1 specific agonists/antagonists on human skin. Abbreviations

ACEA arachidonoyl-chloro-ethanolamide

AEA anandamide

CB1 cannabinoid receptor 1

ECS endocannabinoid system

HF hair follicle

KC keratinocyte

The authors thank Motoko Sugawara for her excellent technical assistance, and Dr. Stephan Tiede for professional advice. The generous professional support of Prof. Masamitsu Ishii and Prof. Hiromi Kobayashi, Osaka, for this work is also gratefully appreciated. Dr. Wolfgang Funk is gratefully acknowledged for harvesting human skin samples for the organ culture.

Additional Information and Declarations

Competing Interests

Author Contributions

Human Ethics

Ralf Paus is an Academic Editor for PeerJ.

Yuval Ramot and Koji Sugawara conceived and designed the experiments, performed the experiments, analyzed the data, wrote the paper.

Nóra Zákány and Balázs I. Tóth performed some of the experiments, and contributed and analyzed qRT-PCR data.

Tamás Bíró contributed and analyzed qRT-PCR data and edited the paper.

Ralf Paus conceived, designed, and supervised the experiments, and wrote the paper.

The following information was supplied relating to ethical approvals (i.e. approving body and any reference numbers):

Institutional review board ethics committee of the University of Luebeck, reference number 06-109.

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
