# Peer review of "A novel control of human keratin expression: cannabinoid receptor 1-mediated signaling down-regulates the expression of keratins K6 and K16 in human keratinocytes in vitro and in situ"

_PeerJ, doi:10.7717/peerj.40_

## Round 0.1 · original submission · Minor Revisions

Dear Dr. Paus,
Please note that, although the suggested changes are "minor revisions", we would like you to carefully take in account the points raised by reviewer 1, and edit the manuscript according to the suggestions of reviewer 2, which will altogether clearly improve the manuscript. Thanks for submitting your work to this journal. I will be looking forward to receive a revised manuscript.

Reviewer 1 ·

Basic reporting

No comment

Experimental design

The authors selected an interesting and important topic. The group is experienced in this issue. There are only some points to be revised.

Methods.
1) The authors used the HaCat cells. According to our experience, these cells do express the CB receptors to a very low extend or even do not express them. Can the authors comment why they used these cells? Why didn´t they use fresh isolated keratinocytes?
2) The authors describe the electronic determination of the staining intensity of immunostainings. This method is critical and vulnerable to methodological variables. How did the authors rule out staining biases and ensure quality? Did the authors stain all sections investigated by the same technician and same day?
3) The authors describe in their methods the qRT-PCR technique but provide only one figure (Fig. 2c). Instead of presentation of the semiquantitative immunostainings, it would be preferable to have a confirmation of their results on RNA or even protein (western blot) level. The authors should also confirm the presence of CB receptors on their HaCat line by PCR or WB.

Validity of the findings

No comments

Comments for the author

No comment

Reviewer 2 ·

Basic reporting

The manuscript is clear, with sufficient explanations in the different sections ; some corrections however have to be done : as reported below, some long sentences should be modified ; some abbreviations are not in their good position. The results are well described in the text and in the figures (some minor modifications are suggested below).

Experimental design

The research presented here concerns the role of cannabinoids in epidermis homeostasis with the selective question of their effects via the CB receptor 1 on the expresssion of hyperproliferative cytokeratins, because of their potential use as therapeutics for hyperproliferaitve skin lesions such as psoriasis. The authors use two experimental models : normal skin organ culture and culture of a keratinocyte line used as a model of hyerproliferative epidermis. They study the effects of CB1 agonists on K6-K16 expression mainly using immunofluorescence.
The introduction defines well the question, although some sentences are too long, so that the reader is somewhat confused at the end of the sentence ( see first paragraph of the introduction for an exemple) ; these long sentences must be modified.
The experimental methods used are standard methods referring to previously published methods, either by the authors or by other groups. Some informations should however be added : what was the localisation of the skin samples in order to well define the type of skin ? if the information is available to the authors, they should precise whether the skin was sun exposed or not ; for the antibodies, the product number should be added ; for LSAB K16 detection, the end of the technique is not mentionned (which chromogen ?)
It is not very clear how the means were calculated : what was considered as a value ? the intensity in one section, the mean intensity of the different sections of one skin sample of one patient ? n should be given in the legend of the figures.
Concerning the figures : the authors must use a same template for the presentation of all their figures : either the titles on the figures are above or under the figures (see fig 2) ; they should choose either small letters or capitals for the numbering of the figures (in the legend and on the figures) ; in the legend 1d should be written as 1-day.

Validity of the findings

The authors show a clear effect of CB1 agonists on the expression of K6-K16 in both type of cultures ; some points need however to be clarified :
K6-K16 are considered as poorly expressed in normal kin ; in the figures presented here, there is a high expression in skin organ culture ; this arises the following question : is the expression of the control samples after one day culture similar to that of in vivo skin ? or is there a modification (namely an increase induced by the culture itself ) ? this point must be clarified because the authors present the two types of culture as two different models, but this does not appear clearly when considering the intensity of keratins expression of controls as presented in the figures.
The authors show a decrease in proliferation via CB1 as previously observed in other conditions by some authors. They conclude that the down-regulation of K6 could result from both direct and indirect (via the decrease of proliferation) effects: the direct effect of agonists is deduced from the observation of a colocalisation of CB1 and K6 ; indeed, in the experimental condition of 1 day culture, a decrease in the superficial keratinocytes could hardly result from an early modification of the differentiation program of keratinization ; however, there is no demonstration of a direct effect of the decrease of proliferation on the expression of K6 ; the authors only observe that there is a correlation between the two effects ; they should clarify their conclusion, last sentence page 12.
Concerning the discussion, it is not sufficiently focused on the main result : down-regulation of K6-K16 : this is only one effect of CB1 agonist on the epidermis, effect which has to be integrated in the whole program of cornifying differentiation of the keratinocytes ; indeed, such a result encourages to further explore the role of cannabinoids in a therapeuthic perspective for the epidermis as well as for other cell types, but the present results do not give sufficient material for a long discussion on that point ; the two last paragraphs can be shortened.
In the conclusion, the first sentence is too general : other groups showed an effect on keratins expression, although not on K6-K6 ; it should be modified as well as the last sentence of the abstract.

Comments for the author

no comments

---

## Round 0.2 · accepted · Accept

I appreciate the careful revision of the manuscript and the detailed responses to the reviewers comments.